# Effects of sequential feeding with adjustments to dietary amino acid concentration according to the circadian rhythm on the performance, body composition, and nutrient balance of growing-finishing pigs

**Alini Mari Veira**[1], **Luan Sousa dos Santos**[2], **Paulo Henrique Reis Furtado Campos**[3], **Danilo Alves Marçal**[1], **Alícia Zem Fraga**[1], **Luciano Hauschild**[1] *

1 Department of Animal Science, School of Agricultural and Veterinarian Sciences, São Paulo State University, Jaboticabal, São Paulo, Brazil, 2 Department of Animal Nutrition and Pastures, Institute of Animal Science, Federal Rural University of Rio de Janeiro, Seropédica, Rio de Janeiro, Brazil, 3 Department of Animal Science, Universidade Federal de Viçosa, Viçosa, Minas Gerais, Brazil

* luciano.hauschild@unesp.br

**Data Availability Statement:** All relevant data are within the paper and its Supporting Information files.

## Abstract

This study aimed to evaluate the effects of a sequential feeding program (SEQ) with diets varying in amino acid (AA) concentrations in the first and last 12 h of the day on the performance, body composition, and nutrient balance of growing-finishing pigs. Sixty-eight castrated male pigs were distributed in four treatments: a daily feeding program (DP) and three SEQs. In the DP, dietary requirements of AA were adjusted daily. In the SEQ, dietary daily requirements of AA were adjusted every 12 h, providing a low AA concentration in period 1 (P1; 00:00–11:59 h) and a high AA concentration in period 2 (P2; 12:00–23:59 h). In the SEQ, three different levels of low and high AA concentrations were evaluated: ±20%, ±30%, and ±40%. The experiment lasted 82 days and was divided into phase 1 (25–50 kg body weight; BW), phase 2 (50–70 kg BW), and phase 3 (70–100 kg BW). During phase 1, irrespective of dietary AA concentration, SEQ pigs had higher lysine intake, protein gain, and phosphorus efficiency than DP pigs (P ≤ 0.05). Pigs in the SEQ showed a tendency for greater average daily gain, body protein, and body lipids compared to the DP pigs (P ≤ 0.10). During phase 2, SEQ pigs showed a tendency for higher average feed intake in P2 compared to DP pigs (P = 0.07); consequently, average daily gain, body protein, and phosphorus retention tended to increase (P ≤ 0.10). During phase 3, SEQ pigs had a higher average feed intake in P2 than DP pigs (P = 0.03). However, they had a similar body composition (P > 0.05) and a tendency for higher nitrogen excretion (P = 0.06) than DP pigs. Our results suggest that SEQ is an effective approach for improving the performance and body composition of growing pigs.

**Funding:** This research was funded by the São Paulo Research Foundation – FAPESP (grant number: 2018/15559-7, São Paulo, Brazil). Alini Mari Veira was supported by a scholarship from the São Paulo Research Foundation – FAPESP (grant number: 2017/18734-1, São Paulo, Brazil) and the National Council for Scientific and Technological Development – CNPq (grant number: 141289/2017-1, Brasília, Brazil). Luciano Hauschild thanks to the scholarship granted from National Council for Scientific and Technological Development – CNPq (grant number: 311054/2020-0, Brasília, Brazil). The funders had no role in study design, data collection and analysis, decision to publish, or preparation of the manuscript.

**Competing interests:** The authors have declared that no competing interests exist.

## Introduction

In swine nutrition programs, diets are predominantly formulated according to animal growth phases, and a single diet is provided to animals during an entire phase (phase feeding programs). However, nutritional requirements may change daily as a result of the dynamics of animal growth [1]. In this context, daily feeding programs may be an effective approach to adjust diets to the daily nutritional requirements of animals, thereby reducing costs and environmental impacts [2, 3]. However, daily feeding does not consider the variations in voluntary feed intake and nutrient metabolism that occur during the 24-hour-day cycle. Therefore, feeding animals according to their physiological and metabolic status within the day could be a next step towards more accurate and environmentally friendly swine nutrition.

Several aspects of animal physiology, including motility of the gastrointestinal tract, nutrient metabolism, and endocrine status, are influenced by the circadian rhythm with decreased or increased frequencies and/or intensities according to the time of day [4]. The pattern of the metabolic state may vary throughout the day, which can be higher in the morning, inducing a catabolic state, and lower in the afternoon, inducing an anabolic state [5]. In addition, studies have shown that feed intake is increased slightly before lights are turned off [6–8]. In this regard, it has been noticed that varying diet protein content throughout the day improved protein metabolism [9], suggesting that the adjustment of the diet nutrient level to the metabolic state can result in benefits for commercial pig production [10].

Considering the differences in the metabolic state throughout the day, sequential feeding consists of providing diets with different nutritional compositions over a given period [11]. This method allows the provision of diets adjusted to the catabolic and anabolic states within the 24-h period as determined by the circadian rhythm. Thus, the hypothesis of the present study was that the adjustment of diets according to the 24-h circadian rhythm, providing low and high concentrations of AA in the period of catabolic and anabolic states, respectively, would improve pig performance, body composition, and nutrient utilization. Therefore, we evaluated the effects of a sequential feeding program varying dietary AA concentrations in the first and last 12 h of the day on the performance, body composition, and nutrient balance of growing and finishing pigs.

## Materials and methods

The experimental protocol was evaluated and approved by the Animal Use Ethics Committee of São Paulo State University, SP, Brazil (protocol number: 012570/18).

### Animals, housing, and experimental design

The study was conducted at the Swine Studies Laboratory of São Paulo State University, Brazil. Sixty-eight castrated male pigs (Agroceres PIC, Rio Claro, SP, Brazil) were housed in a single pen in a negative pressure shed with a 95-m$^2$ concrete floor. The temperature-controlled room was equipped with an evaporative pad cooling system (Big Dutchman, Araraquara, SP, Brazil). The photoperiod was fixed in 12 h of artificial light (06:00–18:00 h; three LED lamps: 240 cm length, 75 W, 6400 luminous flux) and 12 h of darkness (18:00–06:00 h). Water and feed were supplied *ad libitum* using beat-ball drinkers and five automatic intelligent precision feeders (AIPF; University of Lleida, Lleida, Spain).

All animals were equipped with a transponder (plastic button tag containing passive transponders of radio frequency identification; Allflex, Joinville, SC, Brazil) inserted in the right ear of each pig using specific tagger pliers, with an exclusive code that allowed the feeders to identify and record data for each pig. Feeders were described by Pomar, López [12] and comprised a physical component and a logical component; the former was a single-space feeder that

simultaneously blended volumetric amounts of feed stored in independent containers located in the upper part of the AIPF, whereas the latter consisted of software that allowed the programming and registering of all commands executed by the feeders. The AIPF identified each pig as its head entering the feeder and then provided the diet according to the assigned experimental feeds (see next section) in response to each animal request. This allowed the animals to be housed in the same pen, and any pig could access any of the five feeders and receive the prescribed diet according to its treatment. Pigs were adapted to the automatic feeders five days before the trial.

Pigs started the trial with 25 ± 2.67 kg of body weight (BW) and were assigned based on BW to a randomized block design in three blocks: block one with five pigs and blocks two and three with six pigs each. Animals were then distributed into four treatments: a daily feeding program (DP) and three sequential feeding programs (SEQ). In the DP, the dietary concentration of AA was adjusted daily, providing 100% of standardized ileal digestible (SID) AA requirements for the 24-h period. In the SEQ, the dietary daily concentration of SID AA was adjusted every 12 h, providing a low AA concentration from 00:00–11:59 h (period 1; P1) and a high AA concentration from 12:00–23:59 h (period 2; P2). Low and high AA concentrations of SEQ diets corresponded to ±20%, ±30% and ±40% of the SID AA daily requirements. The experimental unit was the individual pig, and each treatment had 17 replicates. The trial lasted 82 days and was divided into three phases: phase 1 (25–50 kg BW), phase 2 (50–70 kg BW), and phase 3 (70–100 kg BW).

## Nutritional requirements, diets, and feeding programs

The NRC model [13] was used to estimate the daily lysine (Lys) requirement curve for pigs from 25 to 100 kg of BW and to formulate the experimental feeds (A and B; Table 1). The protein deposition potential for barrows used as input was integrated into the NRC model (135 g/d). The feed formulation was performed using each ingredient's SID AA content, which was obtained by ingredient analysis (corn and soybean meal) and the SID AA value in Rostagno, Albino [14]. Experimental feeds were formulated to contain similar net energy concentrations and similar AA ratios.

Feed A was formulated with high nutrient concentrations to satisfy the requirements for minerals, vitamins, and energy at the beginning of the growing phase (25 kg BW). Feed B was formulated with low nutrient concentrations to satisfy the requirements for minerals, vitamins, and energy at the end of the finishing phase (100 kg BW). Additionally, feed A was formulated to provide 140% of the ideal AA concentrations, and feed B was formulated to provide 60% of the ideal AA concentrations. The feeds were pelleted at a thickness of 5 mm.

Dietary treatments were obtained by blending feeds A and B in the proportions necessary to obtain the four treatments. The DP contained 100% of the required daily SID AA concentration. The three SEQ feeding programs (SEQ$_{-20/+20}$, SEQ$_{-30/+30}$, and SEQ$_{-40/+40}$) contained different levels of variation in AA concentration every 12 h: in P1, the AA concentrations were −20%, −30%, and −40% of the required daily concentration of SID AA, respectively; in P2, the AA concentrations were +20%, +30%, and +40% of the required daily concentration of SID AA, respectively. This means that pigs from SEQ$_{-20/+20}$, SEQ$_{-30/+30}$, and SEQ$_{-40/+40}$ received diets with 80%, 70%, and 60% of the required daily concentration of SID AA during P1 and 120%, 130%, and 140% of the required daily concentration of SID AA during P2, respectively. These feeding programs were defined as having a graded decrease (P1) or increase (P2) in dietary AA levels between treatments [0 (DP), 20%, 30%, 40%].

## Analytical procedures

Representative samples of corn and soybean meal were collected before the formulation of the feed. These samples were subjected to near-infrared spectroscopy (NIRS; TANGO-FT-NIR

**Table 1. Centesimal and nutritional composition of feeds A and B.**

| Item | Feed A (High AA concentration) | Feed B (Low AA concentration) |
|---|---|---|
| **Ingredient composition as-fed basis, %** | | |
| Corn grain | 64.50 | 91.00 |
| Soybean meal | 30.00 | - |
| Soybean oil | 0.20 | 0.80 |
| Dicalcium phosphate | 1.25 | 1.00 |
| Calcitic limestone | 0.60 | 0.55 |
| Vitamin and mineral premix[1] | 0.20 | 0.20 |
| Salt | 0.20 | 0.20 |
| L-Lysine sulfate (54.6%) | 1.02 | 0.37 |
| L-Threonine | 0.24 | 0.01 |
| DL-Methionine | 0.27 | - |
| L-Tryptophan | 0.05 | 0.02 |
| L-Valine | 0.12 | - |
| Kaolin | 1.29 | 5.79 |
| Choline chloride (60%) | 0.06 | 0.06 |
| **Nutritional composition, %** | | |
| Net energy[2], kcal/kg | 2472 | 2555 |
| Crude Protein[3] | 19.91 | 9.60 |
| Starch[4] | 39.99 | 56.40 |
| Available Phosphorus[2] | 0.32 | 0.22 |
| Calcium[2] | 0.62 | 0.48 |
| Chlorine[2] | 0.17 | 0.17 |
| Sodium[2] | 0.10 | 0.10 |
| SID Lysine[5] | 1.34 | 0.44 |
| SID Threonine[5] | 0.80 | 0.28 |
| SID Methionine[5] | 0.50 | 0.17 |
| SID Methionine + cysteine[5] | 0.77 | 0.32 |
| SID Tryptophan[6] | 0.25 | 0.07 |
| SID Valine[4] | 0.88 | 0.38 |

[1] Mineral and vitamin supplement (per kg of feed): Vit. A (5250 IU), Vit. D3 (750 IU), Vit. E (11 IU), Vit. K3 (1.5 mg), Vit. B1 (1 mg), Vit. B2 (2.4 mg), Vit. B6 (1 mg), niacin (30 mg), B.C. pantothenic (8.1 mg), folic acid (0.53 mg), biotin (0.05 mg), Vit. B12 (16.5 mcg), copper (13.5 mg), iodine (0.19 mg), manganese (37.5 mg), selenium (0.15 mg), zinc (72 mg), iron (72 mg), and cobalt (0.19 mg).

[2] Calculated value.

[3] Crude protein as N × 6.25 according to AOAC [15].

[4] Obtained by polarimetric method [16].

[5] Standardized ileal digestible (SID) amino acid concentrations were calculated using the total amino acid content of feeds A and B obtained by high-performance liquid chromatography (HPLC) and the coefficients proposed by Rostagno, Albino [14].

[6] Standardized ileal digestible (SID) tryptophan concentration values were estimated from analyzed raw matter composition (corn and soybean meal) using the coefficients proposed by Rostagno, Albino [14].

spectrometer, Bruker Optics, Ettlingen, Germany) analysis to determine the total AA values. From this, the SID AA coefficient of corn and soybean meal proposed by Rostagno, Albino [14] was used to calculate the SID AA values that were used in feed formulations (see previous section). Representative samples of feeds A and B were collected throughout the experiment and later homogenized and analysed. The feed was analysed for dry matter (method 930.15) and crude protein (CP) as N × 6.25 (method 968.06; LECO Model F528 N, Leco Corp., St. Joseph City, MI, US) according to AOAC [15]. Starch was obtained by the polarimetric

method [16]. Standardized ileal digestible tryptophan concentration values were estimated from the analysed raw matter composition (corn and soybean meal) using the coefficients proposed by Rostagno, Albino [14]. The total AA composition (except tryptophan) of the feeds was determined by high-performance liquid chromatography (HPLC) based on the chromatography method described by White, Hart [17]. From this, the SID AA coefficient proposed by Rostagno, Albino [14] was used to calculate the SID AA values.

## Experimental measurements

The information of each visit to the feeders, such as pig time of entry and exit and feed intake, was continuously registered using AIPF software. The pigs were weighed on the first day and at the end of each experimental phase (Days 0, 28, 54, and 82). Performance was evaluated based on the average daily feed intake (ADFI; kg/day), average feed intake on P1 (i.e., 00:00–11:59 h; AFI P1; kg/period), average feed intake on P2 (i.e., 12:00–23:59 h; AFI P2; kg/period), SID Lys intake (g/day), average daily gain (ADG; kg/day), gain:feed ratio (G:F; kg/kg), and final BW (kg) in each phase. On Days 0, 28, 54, and 82, total body fat (kg), lean (kg), and bone mineral content (kg) were measured using dual energy X-ray absorptiometry (DXA; Hologic Discovery Instrument, Hologic, Inc., Bedford, MA, USA).

To contain the pigs for total body scanning, each pig was fasted for eight hours prior to receiving anaesthesia. The anaesthesia protocol consisted of a tranquilizer (acepromazine), a sedative (xylazine hydrochloride), and a general anaesthetic (ketamine hydrochloride) administered by intramuscular injection. The concentration of acepromazine was 0.1 mg/kg in all measurements, but those of xylazine hydrochloride and ketamine hydrochloride changed with animal growth as follows: at 25 kg BW, 1.5 mg/kg and 15 mg/kg; at 50 and 70 kg BW, 1.0 mg/kg and 10 mg/kg; and at 100 kg BW, 0.8 mg/kg and 7 mg/kg, respectively. After the scan, pig temperature and respiratory rate were monitored until they recovered completely from anaesthesia.

## Calculations and statistical analysis

Average daily weight gain was calculated as the difference between weights measured at the beginning and at the end of each phase divided by the number of days. Feed intake was calculated using feeding information of each pig registered using AIPF software. From AFI P1 and AFI P2, SID Lys intake, CP intake, and total P intake were calculated considering the amount of nutrients provided by each mixed feed.

Body lean and fat masses obtained by total body scan were converted to body protein (kg) and lipid (kg) following the methodology of Pomar and Rivest [18]. Protein and lipid gain (g/day) were calculated as the difference between the initial and final values in each experimental phase, divided by the number of days. Total body P value was estimated according to Letourneau-Montminy, Narcy [19]. Nitrogen (N) and phosphorus (P) excretion values were obtained by subtracting the respective nutrient retention and intake values. Nitrogen and P efficiencies were calculated by the relation between deposition and intake, respectively.

The homogeneity of the variances and the assumption of a normal distribution of variables were verified using the Box–Cox and Cramer–von Mises tests, respectively. Data were analysed as a randomized complete block design with a fixed effect of feeding programs and random effects of blocks. All analyses were performed using the SAS MIXED procedure (version 9.3; SAS Institute Inc., Cary, NC, USA). The GROUP statement was used to equalize the variances between the means based on the covariance parameter estimates and AIC values. Orthogonal-polynomial contrast analysis was used to compare the daily feeding program with sequential feeding programs (DP vs. SEQ) and to determine the linear and quadratic responses according to increased dietary AA level variation over the 24-hour day. Regarding the levels of variation

in dietary AA, the decrease and increase in P1 and P2, respectively, were similar; therefore, linear and quadratic analyses were performed considering only numbers without positive or negative signals [0 (DP), 20%, 30%, 40%]. Hence, three contrasts were constructed: DP vs. SEQ programs, linear effects (Lin), and quadratic effects (Quad). Statistical differences were considered significant at $P \leq 0.05$, while $0.05 < P \leq 0.10$ was considered a tendency.

## Results

The room temperature averaged $22 \pm 1.59°C$ in phase 1, $21 \pm 1.76°C$ in phase 2, and $20 \pm 1.84°C$ in phase 3. Five pigs (two from $SEQ_{-30/+30}$ and three from $SEQ_{-40/+40}$) were removed from the trial for presenting clinical signs unrelated to the experiment. Pigs had ADFI, ADG, and body composition according to the expected genetic lineage. The AA content was, on average, 8% and 4% below and above the expected for feeds A and B, respectively.

### Performance

During phase 1, SEQ pigs had a tendency for higher ADFI (1.41 vs. 1.29 kg/day; P = 0.10; Table 2), ADG (0.70 vs. 0.63 kg/day; P = 0.06), and BW (44.56 vs. 42.37 kg; P = 0.09) and had higher SID Lys intake (13.6 vs. 12.0 g/day; P = 0.03) than the DP pigs. The ADFI, AFI P2, SID Lys intake, and BW tended ($P \leq 0.09$) to increase in a quadratic manner as the variation in dietary AA levels increased over the 24-h day.

During phase 2, SEQ pigs presented a tendency for higher AFI P2 (1.10 vs. 0.94 kg/period; P = 0.07), ADG (0.83 vs. 0.74 kg/day; P = 0.10), and BW (65.78 vs. 61.43 kg; P = 0.10) than the DP pigs. Moreover, ADG and BW tended to increase linearly ($P \leq 0.09$) as the variation in dietary AA levels increased over the 24-h day. The G:F ratio increased linearly (P = 0.01) as the variation in dietary AA levels increased over the 24-hour day.

During phase 3, SEQ pigs had higher AFI P2 than the DP pigs (1.42 vs. 1.21 kg/period; P = 0.03). Considering global performance, SEQ pigs showed a tendency for higher AFI P2 (1.10 vs. 0.96 kg/period; P = 0.07) and SID Lys intake (16.3 vs. 14.8 g/day; P = 0.08) than the DP pigs.

### Body composition

During phase 1, the SEQ pigs had higher bone mineral content (0.76 vs. 0.71 kg; P = 0.03; Table 3) and protein gain (120.0 vs. 107.4 g/day; P = 0.03) than the DP pigs. Moreover, SEQ pigs showed a tendency for higher body protein (6.44 vs. 6.05 kg; P = 0.06) and tended to respond in a quadratic manner (P = 0.06) as the variation in dietary AA levels increased over the 24-h day. Compared to the DP pigs, the SEQ pigs showed a tendency for higher body lipid content (10.03 vs. 9.72 kg; P = 0.09) and lipid gain (145.5 vs. 125.1 kg/day; P = 0.08), and body lipids increased quadratically (P = 0.03) as the variation in dietary AA levels increased over the 24-h day.

During phase 2, the SEQ pigs showed a tendency for higher body protein (9.84 vs. 9.19 kg; P = 0.10) compared to the DP pigs, and body protein content and protein gain tended to increase linearly ($P \leq 0.10$) as the variation in dietary AA levels increased over the 24-h day. The SEQ pigs had a higher bone mineral content (1.14 vs. 1.05 kg; P = 0.04) than the DP pigs. In phase 3, the SEQ pigs showed a tendency for a higher bone mineral content than the DP pigs (1.64 vs. 1.53 kg; P = 0.09).

### Nutrient balance

During phase 1, the SEQ pigs had higher CP intake than the DP pigs (220.7 vs. 197.3 g/day; P = 0.05; Table 4) and showed a tendency for higher N retention (19.0 vs. 17.2 g/day; P = 0.06)

**Table 2. Performance of pigs in a daily feeding program (DP) or in a sequential feeding program (SEQ) with different levels of variation in amino acid (AA) concentration throughout the day[1].**

| Variables[2] | Variation in recommended amino acid levels in P1 and P2, % | | | | | P-values | | |
|---|---|---|---|---|---|---|---|---|
| | DP | SEQ[3] | | | | | Levels[4] | |
| | 0/0 | −20/+20 | −30/+30 | −40/+40 | SEM | DP vs. SEQ | Lin | Quad |
| **Initial Conditions** | | | | | | | | |
| BW, kg | 25.29 | 25.20 | 25.90 | 25.26 | 0.55 | 0.79 | 0.93 | 0.32 |
| **Phase 1 (25–50 kg BW)** | | | | | | | | |
| ADFI, kg/day | 1.29 | 1.32 | 1.50 | 1.40 | 0.06 | 0.10 | 0.41 | 0.08 |
| AFI P1, kg/period | 0.59 | 0.61 | 0.65 | 0.65 | 0.04 | 0.31 | 0.48 | 0.71 |
| AFI P2, kg/period | 0.71 | 0.77 | 0.85 | 0.76 | 0.04 | 0.12 | 0.86 | 0.09 |
| SID Lys intake, g/day | 12.0 | 12.7 | 14.5 | 13.6 | 0.66 | 0.03 | 0.32 | 0.08 |
| ADG, kg/day | 0.63 | 0.66 | 0.74 | 0.70 | 0.03 | 0.06 | 0.39 | 0.18 |
| G:F, kg/kg | 0.48 | 0.48 | 0.49 | 0.49 | 0.01 | 0.71 | 0.49 | 0.86 |
| BW, kg | 42.37 | 42.86 | 46.23 | 44.60 | 1.20 | 0.09 | 0.29 | 0.07 |
| **Phase 2 (50–70 kg BW)** | | | | | | | | |
| ADFI, kg/day | 1.88 | 2.00 | 2.09 | 2.10 | 0.11 | 0.18 | 0.31 | 0.62 |
| AFI P1, kg/period | 0.95 | 0.91 | 1.00 | 1.01 | 0.06 | 0.72 | 0.25 | 0.58 |
| AFI P2, kg/period | 0.94 | 1.11 | 1.08 | 1.10 | 0.08 | 0.07 | 0.96 | 0.81 |
| SID Lys intake, g/day | 15.3 | 16.0 | 17.0 | 17.3 | 1.01 | 0.15 | 0.33 | 0.75 |
| ADG, kg/day | 0.74 | 0.76 | 0.83 | 0.90 | 0.05 | 0.10 | 0.08 | 0.84 |
| G:F, kg/kg | 0.39 | 0.39 | 0.40 | 0.42 | 0.01 | 0.20 | 0.01 | 0.93 |
| BW, kg | 61.43 | 62.27 | 66.88 | 68.19 | 2.65 | 0.10 | 0.09 | 0.58 |
| **Phase 3 (70–100 kg BW)** | | | | | | | | |
| ADFI, kg/day | 2.50 | 2.63 | 2.63 | 2.59 | 0.11 | 0.34 | 0.78 | 0.92 |
| AFI P1, kg/period | 1.21 | 1.18 | 1.12 | 1.23 | 0.07 | 0.68 | 0.61 | 0.26 |
| AFI P2, kg/period | 1.21 | 1.43 | 1.49 | 1.34 | 0.11 | 0.03 | 0.49 | 0.33 |
| SID Lys intake, g/day | 17.9 | 19.2 | 19.1 | 18.4 | 0.84 | 0.27 | 0.48 | 0.77 |
| ADG, kg/day | 0.99 | 1.00 | 1.01 | 1.01 | 0.04 | 0.70 | 0.84 | 0.89 |
| G:F, kg/kg | 0.40 | 0.38 | 0.38 | 0.38 | 0.01 | 0.11 | 0.63 | 0.81 |
| BW, kg | 89.02 | 90.66 | 95.44 | 96.68 | 3.79 | 0.18 | 0.23 | 0.68 |
| **Global performance (25–100 kg BW)** | | | | | | | | |
| ADFI, kg/day | 1.86 | 1.96 | 2.06 | 2.03 | 0.09 | 0.15 | 0.57 | 0.58 |
| AFI P1, kg/period | 0.91 | 0.90 | 0.92 | 0.96 | 0.05 | 0.78 | 0.35 | 0.85 |
| AFI P2, kg/period | 0.96 | 1.09 | 1.13 | 1.08 | 0.07 | 0.07 | 0.91 | 0.60 |
| Daily SID Lys intake, g/day | 14.8 | 15.8 | 16.7 | 16.5 | 0.82 | 0.08 | 0.51 | 0.55 |
| ADG, kg/day | 0.78 | 0.81 | 0.85 | 0.87 | 0.04 | 0.17 | 0.29 | 0.78 |
| G:F, kg/kg | 0.42 | 0.41 | 0.41 | 0.42 | 0.01 | 0.64 | 0.26 | 0.82 |

[1] Daily feeding program (DP), in which pigs received a diet with 100% of the required daily concentration of SID AA from 00:00 to 23:59 h; sequential feeding programs (SEQ) with different levels of variation in AA concentration, in which pigs received a diet with decreased (−20%, −30%, −40%) of the required daily concentration of SID AA from 00:00 to 11:59 h (period 1), and increased (+20%, +30%, +40%) from 12:00 to 23:59 h (period 2), respectively. Least square means of 17 pigs per treatment.

[2] BW, body weight; ADFI, average daily feed intake; AFI P1, average feed intake in period 1 from 00:00 to 11:59 h; AFI P2, average feed intake in period 2 from 12:00 to 23:59 h; SID, standardized ileal digestible; Lys, lysine; ADG, average daily gain; G:F, gain:feed ratio.

[3] Levels of variation in AA concentration in relation to the required daily concentration of SID AA [13].

[4] Lin, linear effect; Quad, quadratic effect.

and N excretion (16.6 vs. 14.6 g/day; P = 0.10). The SEQ pigs, compared to the DP pigs, presented a tendency for higher P intake (6.7 vs. 6.1 g/day; P = 0.07) and had higher P retention (2.7 vs. 2.4 g/day; P = 0.01) and P retention efficiency (42.19 vs. 39.18%; P = 0.05). The CP

**Table 3. Body composition of pigs in a daily feeding program (DP) or in a sequential feeding program (SEQ) with different levels of variation in amino acid (AA) concentration throughout the day[1].**

| Variables[2] | Variation in recommended amino acid levels in P1 and P2, % | | | | | | P-values | | |
| --- | --- | --- | --- | --- | --- | --- | --- | --- | --- |
| | DP | SEQ[3] | | | | | | Levels[4] | |
| | 0/0 | −20/+20 | −30/+30 | −40/+40 | SEM | | DP vs. SEQ | Lin | Quad |
| **Initial Conditions** | | | | | | | | | |
| Body protein, kg | 3.17 | 3.14 | 3.27 | 3.19 | 0.09 | | 0.76 | 0.75 | 0.38 |
| Body lipid, kg | 6.38 | 6.37 | 6.50 | 6.35 | 0.11 | | 0.83 | 0.91 | 0.31 |
| Bone mineral content, kg | 0.53 | 0.52 | 0.54 | 0.51 | 0.01 | | 0.64 | 0.76 | 0.12 |
| **Phase 1 (25–50 kg BW)** | | | | | | | | | |
| Body protein, kg | 6.05 | 6.19 | 6.72 | 6.42 | 0.18 | | 0.06 | 0.38 | 0.06 |
| Body lipid, kg | 9.72 | 10.00 | 10.84 | 10.06 | 0.30 | | 0.09 | 0.90 | 0.03 |
| Bone mineral content, kg | 0.71 | 0.75 | 0.77 | 0.75 | 0.01 | | 0.03 | 0.55 | 0.31 |
| Protein gain, g/day | 107.4 | 114.6 | 125.3 | 120.1 | 5.23 | | 0.03 | 0.44 | 0.20 |
| Lipid gain, g/day | 125.1 | 139.9 | 158.8 | 137.8 | 10.45 | | 0.08 | 0.88 | 0.12 |
| **Phase 2 (50–70 kg BW)** | | | | | | | | | |
| Body protein, kg | 9.19 | 9.33 | 9.97 | 10.22 | 0.41 | | 0.10 | 0.10 | 0.67 |
| Body lipid, kg | 14.27 | 14.47 | 15.95 | 15.98 | 0.84 | | 0.16 | 0.17 | 0.44 |
| Bone mineral content, kg | 1.05 | 1.10 | 1.14 | 1.17 | 0.04 | | 0.04 | 0.21 | 0.98 |
| Protein gain, g/day | 120.8 | 124.1 | 132.9 | 143.2 | 8.36 | | 0.14 | 0.09 | 0.94 |
| Lipid gain, g/day | 176.6 | 182.6 | 210.7 | 216.1 | 19.64 | | 0.20 | 0.20 | 0.62 |
| **Phase 3 (70–100 kg BW)** | | | | | | | | | |
| Body protein, kg | 13.40 | 13.43 | 13.98 | 14.30 | 0.55 | | 0.40 | 0.24 | 0.85 |
| Body lipid, kg | 22.79 | 23.71 | 25.64 | 25.13 | 1.38 | | 0.17 | 0.44 | 0.45 |
| Bone mineral content, kg | 1.53 | 1.59 | 1.67 | 1.65 | 0.05 | | 0.09 | 0.45 | 0.47 |
| Protein gain, g/day | 143.7 | 140.6 | 140.4 | 143.9 | 6.87 | | 0.78 | 0.73 | 0.82 |
| Lipid gain, g/day | 299.8 | 327.2 | 348.1 | 329.7 | 23.28 | | 0.16 | 0.93 | 0.47 |
| **Global body composition (25–100 kg BW)** | | | | | | | | | |
| Protein gain, g/day | 126.5 | 127.5 | 132.1 | 136.1 | 6.04 | | 0.41 | 0.29 | 0.96 |
| Lipid gain, g/day | 202.2 | 215.2 | 235.6 | 229.5 | 16.29 | | 0.16 | 0.51 | 0.48 |

[1] Daily feeding program (DP), in which pigs received a diet with 100% of the required daily concentration of SID AA from 00:00 to 23:59 h; sequential feeding programs (SEQ) with different levels of variation in AA concentration, in which pigs received a diet with decreased (−20%, −30%, −40%) of the required daily concentration of SID AA from 00:00 to 11:59 h (period 1), and increased (+20%, +30%, +40%) from 12:00 to 23:59 h (period 2), respectively. Least square means of 17 pigs per treatment.

[2] Body protein and lipid were estimated according to Pomar and Rivest [18] from lean and fat mass measured by dual-energy X-ray absorptiometry measurements (DXA).

[3] Levels of variation in AA concentration in relation to the required daily concentration of SID AA [13].

[4] Lin, linear effect; Quad, quadratic effect.

intake, N excretion, and P intake tended to increase in a quadratic manner ($P \leq 0.08$) as the variation in dietary AA levels increased over the 24-h day. P excretion increased quadratically ($P = 0.05$) as the variation in dietary AA levels increased over the 24-h day.

During phase 2, P retention tended to be higher for SEQ pigs than for DP pigs (4.1 vs. 3.5 g/day; $P = 0.08$). N retention tended to increase linearly ($P = 0.09$) as the variation in dietary AA levels increased over the 24-h day. During phase 3, the SEQ pigs, compared to DP pigs, showed a tendency for higher N excretion (30.9 vs. 27.1 g/day; $P = 0.06$) and had a lower N retention efficiency (43.29 vs. 47.07%; $P = 0.04$). Considering the global nutrient balance, the SEQ pigs showed a tendency for higher CP intake (279.6 vs. 255.3 g/day; $P = 0.10$), N excretion (23.8 vs. 21.2 g/day; $P = 0.10$), and P retention (3.8 vs. 3.5 g/day; $P = 0.09$) than the DP pigs.

**Table 4. Nutrient balance of pigs in a daily feeding program (DP) or in a sequential feeding program (SEQ) with different levels of variation in amino acid (AA) concentration throughout the day[1].**

| Variables[2] | Variation in recommended amino acid levels in P1 and P2, % | | | | | P-values | | |
|---|---|---|---|---|---|---|---|---|
| | DP | SEQ[3] | | | | | Levels[4] | |
| | 0/0 | −20/+20 | −30/+30 | −40/+40 | SEM | DP vs. SEQ | Lin | Quad |
| **Phase 1 (25–50 kg BW)** | | | | | | | | |
| CP intake, g/day | 197.3 | 206.2 | 235.7 | 220.3 | 10.65 | 0.05 | 0.34 | 0.07 |
| N retention, g/day | 17.2 | 17.9 | 19.8 | 19.2 | 0.86 | 0.06 | 0.29 | 0.22 |
| N excretion, g/day | 14.6 | 15.4 | 18.1 | 16.2 | 1.09 | 0.10 | 0.60 | 0.08 |
| P intake, g/day | 6.1 | 6.3 | 7.2 | 6.7 | 0.32 | 0.07 | 0.36 | 0.08 |
| P retention, g/day | 2.4 | 2.6 | 2.8 | 2.8 | 0.14 | 0.01 | 0.30 | 0.51 |
| P excretion, g/day | 3.7 | 3.7 | 4.3 | 3.8 | 0.23 | 0.38 | 0.63 | 0.05 |
| N retention efficiency, % | 53.91 | 52.73 | 52.52 | 54.17 | 1.39 | 0.62 | 0.46 | 0.57 |
| P retention efficiency, % | 39.18 | 42.43 | 40.85 | 43.29 | 1.34 | 0.05 | 0.64 | 0.21 |
| **Phase 2 (50–70 kg BW)** | | | | | | | | |
| CP intake, g/day | 261.2 | 272.3 | 290.9 | 294.8 | 16.87 | 0.16 | 0.32 | 0.70 |
| N retention, g/day | 19.3 | 19.8 | 21.2 | 22.9 | 1.33 | 0.14 | 0.09 | 0.94 |
| N excretion, g/day | 22.8 | 24.1 | 25.5 | 25.0 | 1.67 | 0.24 | 0.68 | 0.63 |
| P intake, g/day | 8.4 | 8.7 | 9.4 | 9.5 | 0.53 | 0.17 | 0.31 | 0.66 |
| P retention, g/day | 3.5 | 3.8 | 4.0 | 4.4 | 0.27 | 0.08 | 0.11 | 0.78 |
| P excretion, g/day | 4.9 | 5.0 | 5.4 | 5.2 | 0.32 | 0.44 | 0.69 | 0.48 |
| N retention efficiency, % | 45.79 | 44.77 | 46.26 | 46.45 | 1.49 | 0.98 | 0.40 | 0.71 |
| P retention efficiency, % | 41.86 | 41.75 | 42.45 | 44.08 | 1.57 | 0.58 | 0.27 | 0.80 |
| **Phase 3 (70–100 kg BW)** | | | | | | | | |
| CP intake, g/day | 320.3 | 341.1 | 339.7 | 330.1 | 15.01 | 0.30 | 0.58 | 0.81 |
| N retention, g/day | 22.9 | 22.5 | 22.4 | 23.0 | 1.10 | 0.78 | 0.73 | 0.82 |
| N excretion, g/day | 27.1 | 31.1 | 31.9 | 29.6 | 1.85 | 0.06 | 0.54 | 0.49 |
| P intake, g/day | 10.8 | 11.4 | 11.4 | 11.1 | 0.51 | 0.37 | 0.64 | 0.86 |
| P retention, g/day | 4.5 | 4.6 | 4.9 | 4.6 | 0.22 | 0.49 | 0.98 | 0.37 |
| P excretion, g/day | 6.0 | 6.5 | 6.5 | 6.4 | 0.39 | 0.25 | 0.82 | 0.97 |
| N retention efficiency, % | 47.07 | 43.54 | 41.96 | 44.36 | 1.66 | 0.04 | 0.71 | 0.31 |
| P retention efficiency, % | 44.10 | 42.33 | 43.14 | 41.76 | 1.66 | 0.35 | 0.80 | 0.58 |
| **Global nutrient balance (25–100 kg BW)** | | | | | | | | |
| CP intake, g/day | 255.3 | 270.5 | 286.1 | 282.4 | 13.94 | 0.10 | 0.52 | 0.55 |
| N retention, g/day | 20.2 | 20.4 | 21.1 | 21.7 | 0.96 | 0.41 | 0.29 | 0.96 |
| N excretion, g/day | 21.2 | 23.2 | 24.9 | 23.4 | 1.45 | 0.10 | 0.90 | 0.35 |
| P intake, g/day | 8.4 | 8.8 | 9.2 | 9.1 | 0.45 | 0.16 | 0.58 | 0.56 |
| P retention, g/day | 3.5 | 3.7 | 3.9 | 3.9 | 0.17 | 0.09 | 0.31 | 0.78 |
| P excretion, g/day | 4.8 | 5.0 | 5.3 | 5.2 | 0.30 | 0.27 | 0.72 | 0.52 |
| N retention efficiency, % | 48.22 | 45.98 | 46.01 | 47.30 | 1.05 | 0.12 | 0.36 | 0.61 |
| P retention efficiency, % | 42.11 | 41.92 | 42.48 | 42.77 | 0.97 | 0.79 | 0.52 | 0.90 |

[1] Daily feeding program (DP), in which pigs received a diet with 100% of the required daily concentration of SID AA from 00:00 to 23:59 h; sequential feeding programs (SEQ) with different levels of variation in AA concentration, in which pigs received a diet with decreased (−20%, −30%, −40%) of the required daily concentration of SID AA from 00:00 to 11:59 h (period 1), and increased (+20%, +30%, +40%) from 12:00 to 23:59 h (period 2), respectively. Least square means of 17 pigs per treatment.

[2] CP, crude protein; N, nitrogen; P, phosphorus. Body protein was estimated according to Pomar and Rivest [18] from lean mass measured by dual energy X-ray absorptiometry measurements (DXA) and converted to nitrogen. The total body phosphorus value was estimated according to Letourneau-Montminy, Narcy [19]. Nitrogen and phosphorus efficiency were obtained by the relationship between deposition and intake of phosphorus and protein, respectively.

[3] Levels of variation in AA concentration in relation to the required daily concentration of SID AA [13].

[4] Lin, linear effect; Quad, quadratic effect.

## Discussion

### Overview of the sequential feeding system

Animal physiology is controlled by an internal clock that responds to external stimuli (*zeitgebers*) with a 24-h periodicity [20]. The light-dark cycle is considered the most important stimulus because of its influence on hormonal secretion and, consequently, on the catabolic and anabolic states [5]. Higher catabolism and anabolism rates have been observed at the beginning and end of the day, respectively, in pigs fed *ad libitum* [5]. Accordingly, this study was conducted to evaluate sequential feeding with decreased and increased dietary AA concentrations in the first (P1) and last (P2) 12 h of the day. We hypothesized that providing a lower dietary AA concentration in the morning, when the catabolic state is higher (P1), and a higher dietary AA concentration in the afternoon, when the anabolic state is higher (P2), would improve the nutrient efficiency, performance, and body composition of pigs. In the SEQ, the pigs showed greater performance and body composition, whereas N efficiency was not affected. In addition, the SEQ pigs had greater P efficiency at the beginning of the growing-finishing phase (25–50 kg).

### Performance and body composition

The higher ADFI (8%) of the SEQ pigs during phase 1 (25–50 kg) resulted in a higher SID Lys intake (12%) and, consequently, greater ADG (5%) and lipid (6%), protein (6%), and mineral (6%) deposition compared to the DP pigs. Under adequate health and environmental conditions, pigs regulate feed intake according to their energy requirements [21]. Therefore, the greater feed intake of the SEQ pigs might have resulted from their increased nutrient requirements to support their greater growth rate than the DP pigs. In addition, the SEQ pigs had greater lipid deposition than the DP pigs, which might also be associated with their greater feeding level. Lipid deposition in growing pigs shows a sigmoid nonlinear behaviour as a function of age and tends to increase with increasing BW [22–24]. However, the increase in lipid deposition can also be attributed to a higher feed and, consequently, energy intake [25–27]. In our study, the low to high variation in AA levels quadratically increased the ADFI, BW, body protein, and body lipid in phase 1. It should be noted that the SEQ$_{-30/+30}$ and SEQ$_{-40/+40}$ pigs had quite similar values for BW and body protein. Thus, this quadratic response could be related to the fact that the SEQ$_{-30/+30}$ pigs may have had their maximal response reached in this variation in dietary AA levels over the 24-h day. Therefore, variation above 30% in dietary AA levels over the 24-h day no longer improves performance for pigs weighing 25 to 50 kg.

Conversely, during phase 2, the higher AFI P2 associated with higher dietary AA concentration resulted in a linear increase in ADG, BW, G:F, and protein gain as the variation in AA levels increased; however, lipid gain was similar among the treatments. Protein deposition is mainly determined by the genetic potential of lean tissue growth and the proportion of AA and energy ingested [28]. Therefore, the reason for the SEQ pigs showing better protein deposition than the DP pigs (9% higher) may be the higher intake of AA during the period of most intense anabolism, as there is a positive correlation between protein intake and protein synthesis rate [29].

As a consequence of the circadian rhythm characterized by a diurnal active state and nocturnal fasting period, circulating plasma glucose levels are usually lower in the morning than at night [30]. Additionally, insulin sensitivity in relation to glucose metabolism is lower at night than in the morning [31]. This suggests that the highest use of glucose within cells occurs in the morning. This may suggest that providing a diet with more carbohydrates when insulin sensitivity is greater may be advantageous. In contrast, proteins may be better used at the end

of the day during the anabolic state [30]. Although SEQ diets had similar energy levels, the diets contained different proportions of carbohydrates and proteins. The less concentrated diet (feed B) used at a higher proportion during P1 had a higher concentration of carbohydrates and a lower concentration of proteins, whereas diets that contained higher proportions of concentrate diet (feed A) during P2 were the opposite. This indicates that providing a lower dietary AA concentration (more carbohydrates) in the first 12 h (P1) and a higher dietary AA concentration (more protein) in the last 12 h (P2) of the day would allow for a better use of glucose and protein throughout the day.

In our study, the greater feed intake of the SEQ pigs in P2 did not improve weight or protein gain during phase 3 (70–100 kg). This may indicate that animals reached the maximum protein gain in this phase or even in the previous phase. Pigs in SEQ$_{-40/+40}$, for example, showed similar protein gain in phases 2 and 3 (143.2 vs. 143.9 g/day). Therefore, once the maximum protein gain was reached, the increment in AA intake likely resulted in AA deamination due to excess metabolism. As protein deposition does not depend exclusively on nutritional strategy but also on genetic factors, adjusting the dietary AA concentration according to the circadian rhythm of the catabolic and anabolic states can be an effective approach for animals that have not yet reached the maximum protein deposition.

## Nutrient balance

Our results showed that the positive effect of SEQ on ADFI resulted in a tendency for greater N and P retention during phase 1. Moreover, the positive effect of SEQ on feed intake in P2 combined with the AA and P levels increased during this period, resulting in higher N and P retention during phase 2. Thus, in addition to the effect of feed intake, the SEQ pigs ingesting fewer nutrients in P1 and more nutrients in P2 may have resulted in a greater availability of nutrients during the anabolic period. However, although feed intake in P2 was higher for the SEQ pigs during phase 3, the N and P balance retention was worse than that in the DP pigs. Again, this could be explained by the maximum protein deposition that was reached in phase 2 of the experiment.

In our study, the efficiency of N utilization was similar between SEQ and DP, while P utilization was improved for SEQ during phase 1. Skeletal tissues express a strong circadian clock, and numerous studies have shown that bone metabolic functions are regulated in a circadian manner [32, 33]. Thus, P can be considered an essential nutrient that controls circadian function in both skeletal and nonskeletal peripheral tissues [34]. Therefore, adequate intake of this nutrient during the day may improve P retention and the efficiency of P utilization in pigs, as observed in our study.

Overall, the adjustment of dietary AA and P concentrations according to the circadian rhythm (catabolic and anabolic states) may better meet nutritional requirements, allowing animals to fully reach their growth potential. These findings suggest that SEQ is a potential strategy to improve N and P deposition in pigs when compared with standard feeding programs. Additionally, feed represents a large part of the cost of pig production, and the P component represents an important proportion of feed cost [35, 36]. As the SEQ pigs herein showed the best values for P efficiency at the beginning of the growing phase (25–50 kg BW), SEQ may be more economically advantageous than daily feeding and even standard programs.

## Conclusion

In summary, the results of the present study indicate that adjusting the dietary AA concentrations according to the circadian rhythm (catabolic and anabolic states) improves protein deposition and tends to increase ADFI and ADG for pigs weighing 25–50 kg. Moreover, this

adjustment tends to increase ADG, BW, and body protein in pigs weighing 50–70 kg. These responses were observed when the dietary AA levels were 40% lower and higher than the recommended values in the first and last 12 h of the day, respectively. Therefore, the sequential feeding system presented herein, with the adjustment of the diet nutrient concentration according to the circadian rhythm, is an effective approach for improving the performance and body composition of growing pigs.

## Supporting information

**S1 Table. Analyzed nutritional composition of soybean meal and corn.**
(DOCX)

**S2 Table. Performance of the experimental pigs.**
(DOCX)

**S3 Table. Body composition of the experimental pigs.**
(DOCX)

**S4 Table. Nutrient balance of the experimental pigs.**
(DOCX)

## Acknowledgments

The authors would like to thank Prof. Gustavo do Valle Polycarpo (São Paulo State University —UNESP, Dracena Campus, São Paulo, Brazil) for supporting the statistical analysis.

## Author Contributions

**Conceptualization:** Alini Mari Veira, Paulo Henrique Reis Furtado Campos, Luciano Hauschild.

**Formal analysis:** Alini Mari Veira, Luan Sousa dos Santos.

**Funding acquisition:** Luciano Hauschild.

**Investigation:** Alini Mari Veira.

**Project administration:** Luciano Hauschild.

**Supervision:** Luciano Hauschild.

**Writing – original draft:** Alini Mari Veira.

**Writing – review & editing:** Alini Mari Veira, Luan Sousa dos Santos, Paulo Henrique Reis Furtado Campos, Danilo Alves Marçal, Alícia Zem Fraga, Luciano Hauschild.

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
