## [Decision Letter · Decision Letter 0]

13 Oct 2021

PONE-D-21-30142Effects of sequential feeding with adjustments in dietary amino acid concentration according to the circadian rhythm on performance, body composition, and nutrient balance of growing-finishing pigsPLOS ONE

Dear Dr. Veira,

Thank you for submitting your manuscript to PLOS ONE. After careful consideration, we feel that it has merit but does not fully meet PLOS ONE’s publication criteria as it currently stands. Therefore, we invite you to submit a revised version of the manuscript that addresses all the points raised during the review process. Particularly, you will see that additional information regarding your manuscript have been asked by the reviewer.

We look forward to receiving your revised manuscript.

Kind regards,

Francois Blachier, PhD

Academic Editor

PLOS ONE

Journal Requirements:

"This study was funded by the São Paulo Research Foundation – FAPESP (grant number: 2018/15559-7, São Paulo, Brazil). Alini Mari Veira was supported by a scholarship from the São Paulo Research Foundation – FAPESP (grant number: 2017/18734-1, São Paulo, Brazil) and the National Council for Scientific and Technological Development – CNPq (grant number: 141289/2017-1, Brasília, Brazil)."

Reviewers' comments:

Reviewer's Responses to Questions

**Comments to the Author**

1. Is the manuscript technically sound, and do the data support the conclusions?

Reviewer #1: Yes

2. Has the statistical analysis been performed appropriately and rigorously? 

Reviewer #1: Yes

3. Have the authors made all data underlying the findings in their manuscript fully available?

Reviewer #1: Yes

4. Is the manuscript presented in an intelligible fashion and written in standard English?

Reviewer #1: Yes

5. Review Comments to the Author

Reviewer #1: This study was conducted to investigate the effects of a sequential feeding program (SEQ) with diets varying in amino acid (AA) concentrations in the first and last 12 hours of the day on performance, body composition, and nutrient balance of growing-finishing pigs. The results are interesting, yet there still are some problems.

1. It is necessary that the manuscript has a native English speaker to correct the language and grammar.

2. In the Introduction section, I hope the authors could introduce the current status of research in this area.Here are two important references to this topic:

[1] Chunyan Xie, Xin Wu, Jun Li, Zhiyong Fan, Cimin Long, Hongnan Liu, Even Patrick, Francois Blachier, Yin Yulong. Effects of the sequence of isocaloric meals with different protein contents on plasma biochemical indexes in pigs. Plos One, 2015, 10(8):e0125640

[2]Chunyan Xie, Xinyi Duan, Cimin Long, Xin Wu*. Hepatic lipid metabolism is affected by daily 3-meal pattern with varying dietary crude protein with a pig model. Animal nutrition, 2020, 6: 16-23

3. In the Materials and methods section, did the authors evaluate the normality and homogeneity of variances of the data? In addition, the authors said that results were identified as trends when 0.05 < P ≤ 0.10, is this definition scientific and conform to the laws of statistic?

4. Is the feed intake of pigs among four treatments in period 1 and period 2 the same? In addition, how do you ensure that every pig's daily intake of total amino acids is consistent?

6. PLOS authors have the option to publish the peer review history of their article (what does this mean?). If published, this will include your full peer review and any attached files.

Reviewer #1: **Yes: **Xin WU

---

## [Author Response · Author response to Decision Letter 0]

26 Nov 2021

Academic Editor

Journal Requirements:

1) Please ensure that your manuscript meets PLOS ONE's style requirements, including those for file naming. The PLOS ONE style templates can be found at: https://journals.plos.org/plosone/s/file?id=wjVg/PLOSOne_formatting_sample_main_bbod.pdf;
https://journals.plos.org/plosone/s/file?id=ba62/PLOSOne_formatting_sample_title_authors_affiliations.pdf.

Authors: Corrected according to the editor’s suggestion. We removed the short title from title page; We adjusted the font size of the table footnotes; We listed supporting information captions at the end of the manuscript. 

Authors: Checked according to the editor’s suggestion.

3) We note that the grant information you provided in the ‘Funding Information’ and ‘Financial Disclosure’ sections do not match.

Authors: Corrected according to the editor’s suggestion.

4) Thank you for stating the following financial disclosure:

"This study was funded by the São Paulo Research Foundation – FAPESP (grant number: 2018/15559-7, São Paulo, Brazil). Alini Mari Veira was supported by a scholarship from the São Paulo Research Foundation – FAPESP (grant number: 2017/18734-1, São Paulo, Brazil) and the National Council for Scientific and Technological Development – CNPq (grant number: 141289/2017-1, Brasília, Brazil)."

Authors: Additional information was included according to the editor’s suggestion.

5) In your Data Availability statement, you have not specified where the minimal data set underlying the results described in your manuscript can be found. PLOS defines a study's minimal data set as the underlying data used to reach the conclusions drawn in the manuscript and any additional data required to replicate the reported study findings in their entirety. All PLOS journals require that the minimal data set be made fully available. For more information about our data policy, please see

http://journals.plos.org/plosone/s/data-availability.

Upon re-submitting your revised manuscript, please upload your study’s minimal underlying data set as either Supporting Information files or to a stable, public repository and include the relevant URLs, DOIs, or accession numbers within your revised cover letter. For a list of acceptable repositories, please see

http://journals.plos.org/plosone/s/data-availability#loc-recommended-repositories. 

Any potentially identifying patient information must be fully anonymized.

Important: If there are ethical or legal restrictions to sharing your data publicly, please explain these restrictions in detail. Please see our guidelines for more information on what we consider unacceptable restrictions to publicly sharing data:

http://journals.plos.org/plosone/s/data-availability#loc-unacceptable-data-access-restrictions. Note that it is not acceptable for the authors to be the sole named individuals responsible for ensuring data access.

Authors: Additional supporting information was included according to the editor’s suggestion (S1 Table. Analyzed nutritional composition of soybean meal and corn; S2 Table. Performance of the experimental pigs; S3 Table. Body composition of the experimental pigs; S4 Table. Nutrient balance of the experimental pigs).

Review Comments to the Author

Reviewer #1

This study was conducted to investigate the effects of a sequential feeding program (SEQ) with diets varying in amino acid (AA) concentrations in the first and last 12 hours of the day on performance, body composition, and nutrient balance of growing-finishing pigs. The results are interesting, yet there still are some problems.

1) It is necessary that the manuscript has a native English speaker to correct the language and grammar.

Authors: Thank you for your valuable comments. The manuscript was edited for proper English language, grammar, spelling, and overall style (certificate attached). We hope the revised manuscript will meet the requirements of academic publishing in Plos One.

2) In the Introduction section, I hope the authors could introduce the current status of research in this area. Here are two important references to this topic:

[1] Chunyan Xie, Xin Wu, Jun Li, Zhiyong Fan, Cimin Long, Hongnan Liu, Even Patrick, Francois Blachier, Yin Yulong. Effects of the sequence of isocaloric meals with different protein contents on plasma biochemical indexes in pigs. Plos One, 2015, 10(8):e0125640

[2]Chunyan Xie, Xinyi Duan, Cimin Long, Xin Wu*. Hepatic lipid metabolism is affected by daily 3-meal pattern with varying dietary crude protein with a pig model. Animal nutrition, 2020, 6: 16-23

Authors: Additional references have been included according to the reviewer’s suggestion (lines 61-64).

3) In the Materials and methods section, a) did the authors evaluate the normality and homogeneity of variances of the data? b) In addition, the authors said that results were identified as trends when 0.05 <P ≤ 0.10, is this definition scientific and conform to the laws of statistic?

Authors: Thank you referee for your comment. a) Yes. We performed the analysis of normality and homogeneity of variance. We have included this information in the material and methods section (lines 208-209). b) The use of trend in statistical analysis divides opinions, while some statisticians are against this approach, some are in favor in using it. In animal nutrition manuscripts, it has been common to use tendency to make inferences in the results, however care has been taken in the conclusions. In our study, the pigs were housed in a group with all animals in a single pen, which simulate a farm condition. Therefore, it was expected that the variability would be higher, resulting in notable numerical differences, but not always with P < 0.05. So we use trend when 0.05 <P ≤ 0.10. However, some studies consider a trend even when the P-value is a little higher than 0.10. For example in Schiavon et al. (2018), the authors use P = 0.13 as a trend. In our study, in the conclusion section we were clear about trends in order do not confuse readers.

Schiavon S, Dalla Bona M, Carcò G, Carraro L, Bunger L, Gallo L (2018) Effects of feed allowance and indispensable amino acid reduction on feed intake, growth performance and carcass characteristics of growing pigs. PLoS ONE 13(4): e0195645. https://doi.org/10.1371/journal.pone.0195645

4) a) Is the feed intake of pigs among four treatments in period 1 and period 2 the same? b) In addition, how do you ensure that every pig's daily intake of total amino acids is consistent?

Authors: a) No. During phase 2 (50–70 kg BW) and during the global experiment (25–100 kg BW), the SEQ pigs presented a tendency for a higher feed intake in P2 compared to the DP pigs. While during phase 3 (70–100 kg BW), the feed intake in P2 was higher for the SEQ pigs than the DP pigs. The feed intake in P1 did not differ between treatments throughout the experiment. This information is described in the Results section.

b) In fact, we cannot ensure that each pig ingested the exact daily amount of total amino acids. In this study, the pigs were fed ad libitum, so it is expected that the animals may adjust their feed intake in order to meet their nutritional requirements. Once that pigs performed according to genetic company recommendation, we assume that the daily minimal amino acid requirement was met for all the pigs. Another interesting point, is that according to the literature and previous studies conducted in our lab, a higher feed intake in P2 is expected due to metabolic variations. So, we provide a greater amino acid diet level in P2 to test our hypothesis. However, we believe that the animals were not nutritionally deficient in amino acids. Feed intake in P1 (lower AA concentration) did not differ, while in P2 (higher amino acids concentration) the SEQ pigs were able to adjust their feed intake according to their nutritional requirement, consequently presenting a higher feed intake in P2.

---

## [Editor Report · Decision Letter 1]

1 Dec 2021

Effects of sequential feeding with adjustments to dietary amino acid concentration according to the circadian rhythm on the performance, body composition, and nutrient balance of growing-finishing pigs

PONE-D-21-30142R1

Dear Dr. Veira,

We’re pleased to inform you that your manuscript has been judged scientifically suitable for publication and will be formally accepted for publication once it meets all outstanding technical requirements.

Kind regards,

Francois Blachier, PhD

Academic Editor

PLOS ONE
---

## [Editor Report · Acceptance letter]

14 Dec 2021

PONE-D-21-30142R1 

Effects of sequential feeding with adjustments to dietary amino acid concentration according to the circadian rhythm on the performance, body composition, and nutrient balance of growing-finishing pigs 

Dear Dr. Veira:

I'm pleased to inform you that your manuscript has been deemed suitable for publication in PLOS ONE. Congratulations! Your manuscript is now with our production department. 

Kind regards, 

on behalf of

Dr. Francois Blachier 

Academic Editor

PLOS ONE